# Healthy Nutrition and Physical Activity in Childcare: Views from Childcare Managers, Childcare Workers and Parents on Influential Factors

**DOI:** 10.3390/ijerph15122909

**Published:** 2018-12-19

**Authors:** Ilona van de Kolk, Anne J. M. Goossens, Sanne M. P. L. Gerards, Stef P. J. Kremers, Roos M. P. Manders, Jessica S. Gubbels

**Affiliations:** Department of Health Promotion, NUTRIM School for Nutrition and Translational Research in Metabolism, Maastricht University, P.O. Box 616, 6200 MD Maastricht, The Netherlands; ajm.goossens@student.maastrichtuniversity.nl (A.J.M.G.); sanne.gerards@maastrichtuniversity.nl (S.M.P.L.G.); s.kremers@maastrichtuniversity.nl (S.P.J.K.); r.manders@alumni.maastrichtuniversity.nl (R.M.P.M.); jessica.gubbels@maastrichtuniversity.nl (J.S.G.)

**Keywords:** childcare, pre-school, environment, nutrition, physical activity, qualitative, consistency, socio-ecological model, Netherlands, behaviour

## Abstract

Childhood obesity is an important public health issue influenced by both personal and environmental factors. The childcare setting plays an important role in children’s energy balance-related behaviours (EBRB), such as physical activity, sedentary behaviour and healthy nutrition. This study aimed to explore facilitators and barriers of healthy EBRB in childcare in a comprehensive way, from the perspective of three crucial stakeholders: childcare managers, childcare workers and parents. A qualitative study was performed using semi-structured interviews. Content analysis was performed using the ‘Environmental Research framework for weight Gain prevention’ (EnRG framework) to guide the analysis. Forty-eight interviews were held with a total of 65 participants (9 childcare managers, 23 childcare workers and 33 parents). Influential factors in all types of environment (physical, sociocultural, economic and political) were mentioned. Although a need for change was not always expressed, the interviews revealed opportunities for improvement of healthy EBRB in childcare. These opportunities were related to the sociocultural, physical and political environment. Childcare workers and managers expressed an influence of the home setting on the childcare setting, resulting in a need for more congruence between these settings. There are opportunities for improvement in the childcare setting to promote healthy EBRB in young children in the Netherlands. It appears important to align intervention components between the childcare and home setting.

## 1. Introduction

Many children are growing up in an obesogenic environment, resulting in a high intake of energy-dense foods, low levels of physical activity and high levels of sedentary behaviour [1,2,3,4]. These unfavourable energy balance-related behaviours (EBRB) have resulted in an increase in the prevalence of childhood overweight and obesity [5,6]. In the Netherlands, 8.0% of 2-year-old boys and 8.3% of 2-year-old girls are overweight including 0.7% obesity in both boys and girls, and these numbers increase to 9.1% (boys, overweight), 16.3 % (girls, overweight), 1.1% (boys, obese) and 2.6% (girls, obese) for 4-year-old children [5]. These numbers are comparable to the prevalence of overweight and obesity in other Northern European countries, but are fairly favourable compared to the prevalence in other Western countries [7,8]. As the prevalence of childhood overweight and obesity is expected to keep rising, the prevention of childhood overweight and obesity is still an important public health issue [8]. Lifestyle behaviours are developed early in life and are known to track into adulthood [9]. Furthermore, weight status between two and six years of age is most predictive for adult weight status and overweight and obesity-related diseases [10,11]. In practice and research, increased attention has been paid to the prevention of childhood overweight and obesity and the promotion of a healthy lifestyle in young children [12,13,14]. 

The home environment, in particular parents, exerts an important influence on child EBRB [15,16,17,18]. Many children are also cared for in formal childcare such as day care or preschool [19]. In the Netherlands, 41% of children under the age of three attend formal childcare, ranking childcare use in the Netherlands among one of the highest in Europe [20]. This percentage doubles to 82% for children between three and five years old [20]. On average, children attend formal childcare for about seventeen hours per week [21]. Studies have shown that childcare use could result in an increased risk for overweight and obesity in children [22,23,24,25]. Potential factors influencing this increased risk include foods consumed during childcare [26]; limited opportunities to be physically active [24]; and staff behaviours [27]. There is also evidence of a protective role [24]. The use of favourable nutrition- and physical activity-related practices by childcare staff, such as prompting children to be physically active and using non-food rewards for trying new foods, has been shown to be positively associated with the related behaviours in children [28]. Along with these results from quantitative research, a number of studies have examined perceived facilitators and barriers of healthy EBRB in the childcare setting. A recent review describes more general themes in which barriers and facilitators for physical activity and sedentary behaviour are perceived [29]. They include the child, the home, the out-of-home childcare, parent-childcare provider interactions, environmental factors, safety and weather [29]. More specifically, perceived barriers and facilitators in the case of physical activity are lack of indoor or outdoor play space or materials, safety rules, and support of colleagues [30]. Regarding healthy nutrition, lack of policy, lack of training of staff, and budget and time constraints were perceived barriers [31,32]. Thus, various factors within the childcare setting play an important role in the healthy development of young children’s EBRB and weight status. 

The improved understanding of the complexity of childhood overweight and obesity has directed researchers to adopt a socio-ecological perspective, acknowledging the importance of the multi-level (e.g., intrapersonal, interpersonal and community level) influence of determinants [33]. As Sallis et al. described, ‘ecological models are believed to provide comprehensive frameworks for understanding the multiple and interacting determinants of health behaviours’ [34]. The Environmental Research framework for weight Gain prevention (EnRG) adopts the socio-ecological research paradigm and proposes that the environment influences EBRB directly, as well as through the mediation of cognitive variables (attitude, subjective norm and perceived behavioural control) [35]. Further, it proposes the moderating influence of personal and behavioural factors on environment–behaviour relationships [35]. The EnRG framework describes different types of environment, based on the ANGELO framework, namely, sociocultural (what is the social and cultural background), physical (what is available), economic (what are the costs) and political (what are the rules) [35,36]. The aim of this study was to explore facilitators and barriers in childcare to promote healthy nutrition and physical activity, using the EnRG framework and taking the views of different stakeholders (i.e., childcare managers, childcare workers and parents) into account.

## 2. Materials and Methods 

### 2.1. Setting

The study was conducted in the south of the Netherlands. In the Netherlands, there are several different types of formal childcare: centre-based childcare for infants and pre-schoolers (including preschool and centre-based day care), family-based childcare for infants and pre-schoolers, and after-school care for children in primary school [37]. The current study focuses on the first type: formal centre-based childcare, with a specific focus on pre-schoolers. Pre-schools provide half-day childcare with a focus on playful learning to prepare children for primary school. In this type of childcare, there is only one moment during which children consume food (snack time), and they often bring their own food. Children between 2 and 4 years old can attend preschool [37]. Centre-based childcare provides whole-day childcare and usually focuses less on educational goals [37]. In this type of childcare, there are several moments during which children consume food, and the childcare institutions mostly provide the food products. Children 0–4 years old are able to attend centre-based childcare [37]. Parents can receive a general childcare benefit for formal childcare from the government, based on their working hours and income [38]. 

### 2.2. Study Sample and Recruitment

In-depth interviews were held with centre-based childcare managers, pre-school childcare workers, and parents of children attending pre-school. All childcare managers had supervisory and policy-making responsibilities. Childcare workers were responsible for the daily supervision of the children and provision of the educational activities at preschool. Childcare workers should be minimally trained with a lower vocational pedagogical education [39]. All interviews were limited to those working with children aged 2–4 years old. A combination of purposive (childcare workers) and convenience sampling (childcare managers and parents) was used. Childcare managers of fifteen childcare facilities were approached by telephone and asked to participate in an interview; nine managers (from eight facilities, 53.3%) were willing to participate. Reasons for non-participation were lack of time (N = 1) and inability to reach the manager (N = 6). 

Childcare workers of ten pre-schools were asked to participate in the interviews, and all pre-schools participated (100%). They received brief written information about the interview by e-mail, before an appointment was made by telephone. The interviews with the childcare workers were held with those who were present at the time of the interview, mostly two and occasionally three childcare workers per pre-school. A total of twenty-three childcare workers participated in the interviews. To inform the parents, a pamphlet announcing the presence of the researcher was distributed among the pre-schools that agreed to participate. Parents were asked to participate in a short interview during or directly after drop-off and picking-up times at the pre-school. Thirty-three parents agreed to participate. Verbal informed consent from all participants was obtained before conducting the interviews. The Maastricht University Medical Centre+ Medical Ethics Committee reviewed and approved this study as part of a larger research project (METC163022). 

### 2.3. Data Collection Methods

A qualitative research design with semi-structured interviews was used. A comprehensive theoretical framework (EnRG framework [35]) was used to guide the development of the topic list for the interviews. Questions asked during the interviews included: ‘How do you feel about healthy nutrition/physical activity in young children?’ and ‘How do you feel about the space and play materials at this facility?’ Interviews with the childcare managers were conducted by AG, and all other interviews were conducted by IK. Almost all interviews with the childcare managers and workers took place in a quiet environment. Most interviews with the parents took place in a public area near the pre-school and, therefore, were not always quiet places. All interviews were held in Dutch and were audio-recorded (Olympus VN-2100 PC, digital voice recorder or Android Application Smart Voice Recorder). The interviews were held between March 2015 and June 2016. 

### 2.4. Data Processing and Analysis 

All interviews were transcribed verbatim. If words or sentences were unclear, a second researcher was consulted to complete the transcript. All transcripts were anonymized by removing names and locations. A directed content analysis approach was adopted [40]. The EnRG framework [35] was used as the theoretical framework for the analysis. The constructs of this framework (e.g., the types of environments and cognitive determinants) formed the basis of the content analysis. Additionally, codes were used to increase specificity, such as ‘nutrition or physical activity’, ‘indoor- or outdoor- play area’, or ‘influence of other pre-schools or other pre-school teachers’ that arose from the data. For the construct ‘political environment’ content describing rules, regulations, policies regarding nutrition or physical activity in the childcare setting are considered shaping the political environment. Initial analysis was done by AG (childcare managers) and RM (all other interviews). The analysis was checked by IK for consistency of coding with the theoretical framework, and an additional analysis of cognitive variables was done for the interviews with the childcare workers and parents. Data analysis was performed using QSR International’s NVivo 11 qualitative data analysis software (QSR International, Doncaster, Victoria, Australia).

## 3. Results

### 3.1. Respondents 

Forty-eight interviews were held with a total of 65 participants (9 childcare managers, 23 childcare workers and 33 parents). The interviews with childcare managers lasted on average 42 min (range: 32–50), lasted 34 min (range: 21–60) on average with childcare workers, and almost 10 min (range: 4–23) on average with parents. All childcare managers were female, with an average age of 40.1 years, an average working experience as manager of 6.3 years, and 55.5% of the managers had experience as a childcare worker (Table 1). All childcare workers were female, with an average age of 49.8 years and a working experience in childcare of 17.6 years (Table 1). Of the parents, 22.9% was male, 25.0% had low education, 48.4% was employed, and 87.1% was in a relationship (Table 2). 

### 3.2. Cognitive Mediators

#### 3.2.1. Attitude

All respondents (managers, childcare workers and parents) had a positive attitude towards healthy nutrition and physical activity in young children. There were various beliefs underlying this positive attitude. All respondents mentioned the belief that healthy nutrition and physical activity are important for the health of the children. For childcare workers, an important belief regarding physical activity was to give the children the opportunity to go outside when they do not have this opportunity at home. *‘We find it especially important because we know a lot of the children live in flats, do not have a garden and do not go outside often’ (CW3.2)*. The respondents believed that being active made children happy and that children enjoyed being physically active (see Appendix A for additional quotes). 

With regard to healthy nutrition, childcare managers believed that it was very important to encourage children to eat as healthily as possible. However, some childcare managers did not think that this meant that all unhealthy foods should be banned. *‘I hear in some organisations that they ban juice completely, and then I think: Come on! … I think it is important for children to learn to drink water, but to not give juice at all..., that is not what puts on weight’ (CM1).* Childcare workers and parents both mentioned this belief as well, in particular with regard to sugar-sweetened beverages. Parents believed that it was important to keep a balance. It was often mentioned that they did not really mind their children eating sweets or snacks or drinking sugar-sweetened beverages, as long as it was limited to a little bit and balanced with healthy products. *‘I do not think lemonade is really necessary … but I do not mind it for one time a day’ (P16).*


#### 3.2.2. Social Norm

Childcare workers and managers did not mention many social influences on how they handle nutrition and physical activity. One childcare manager even said, *‘If you do not see the value of what we are doing here, then you may not be a parent that fits here’ (CM4).* On the other hand, parents were often mentioned as an important influence. In particular, in relation to birthday treats, they experienced that there is still a widespread preference of parents to provide sweets or snacks instead of a healthy treat, because it is a festive occasion. *‘Well, we could try it, but then I think if the switch to fruit as a snack is so difficult, then the birthday treat will be… I think for most families you will really step on their toes’ (CW2.1)*. Childcare workers found it difficult to address this belief with parents, and often capitulated by allowing unhealthy treats. Childcare managers also talked about the more general focus of society on healthy nutrition and physical activity. They experienced that this also forced them to be conscious about it. *‘You indeed notice that more and more parents ask questions or say, “I do not want them to participate with birthday treats.” Parents are very, very occupied with it and, therefore, we are also very occupied with it’ (CM5)*. 

#### 3.2.3. Perceived Behavioural Control

All respondents commonly expressed that it is ‘in the nature’ of children to be physically active, and therefore, they did not perceive it as difficult to ensure that the children would be physically active. However, some childcare workers expressed that they felt incapable and insecure about having the children in a physical education room. *‘That’s just not for me, my nerves were in tatters, they were climbing in everything and they are so little, so you think oh if they fall out of it! … and then with just the two of us, that was impossible’ (CW10.2)*. With regard to healthy nutrition, all respondents felt they were capable of promoting healthy nutrition in children. Child preferences or dislike of certain foods were often mentioned, but none of the childcare workers expressed that this hindered their ability to promote healthy nutrition for these children. 

### 3.3. Environmental Facilitators and Barriers in Childcare

#### 3.3.1. Physical Environment 

The respondents emphasized the importance of the availability of healthy food products in childcare, such as fruit and vegetables, healthy spreads for sandwiches, and drinks without sugar or low in sugar. All facilities, except for two day-care centres, served lemonade to the children, and although this may not be regarded as a healthy choice, almost all respondents approved or tolerated the serving of lemonade in childcare. *‘I don’t think it’s necessary to switch to water, they only get a little to drink, and we use a small amount of lemonade’ (CW6.1)*. This was sometimes also the result of not knowing a healthier alternative than lemonade. *‘Yes, lemonade we are aware of it, but what do you give them otherwise? Juice mixed with water?’ (CW4.1)*. In pre-schools, most respondents were very positive about the availability and variety of the fruits brought by parents. However, there were childcare workers who experienced that parents would often bring fruits that are on sale or no fruits at all, and vegetables were only sporadically available, reducing variability. In centre-based day care centres, the availability of food products was predominantly determined by rules and regulations. All childcare managers took them into account to ensure a healthy food environment for the children. However, energy-dense snacks such as cookies were more often available in these childcare institutions compared to pre-schools. 

With regard to physical activity, parents and managers in particular were satisfied with the possibilities. In most facilities there was an outdoor playing area, which was suitable for children of that age group. Often an indoor play area was also present. Parents and childcare managers were positive about the availability of play materials, both indoors and outdoors, found them age-appropriate and also stimulating for motoric development. *‘They have big playing areas, also inside … They can climb, they can slide, they can bike, and they can run … all that a child should be able to do, they can do here’ (P8)*. With regard to play materials, parents often referred positively to puzzles, painting materials and building blocks, which may be more related to sedentary activities. Childcare workers were more critical about the physical environment with regard to physical activity, compared to the other respondents. In particular, they experienced lack of safety, lack of challenging play materials and, more generally, the appearance and accessibility of the outdoor playground as barriers to be physically active with the children. *‘There is an outdoor playing area with a sandpit and ‘rolling materials’, so that’s what we have. But the sandpit is small with a high border, which isn’t safe … there is totally nothing green or grass … They can’t climb, all they can do is ride a bike or scooter, that’s it’ (CW8.2)*. Furthermore, a lack of availability of an indoor physical education room and time, especially the high demands on the available time that they have with the children, were important barriers mentioned by the childcare workers. Several of them expressed a need for greater variety in play equipment to be able to promote different locomotor skills (e.g., climbing and balancing). They felt that with their current offer, they were unable to stimulate the development of these skills in children. Many childcare workers mentioned that they would like to have more natural elements in the outdoor playground, such as grass, a little garden or hills to climb on, *‘… I would say: make something with grass there, then you also have something hillier they* [the children] *can climb on’ (CW3.1)*.

#### 3.3.2. Sociocultural Environment

Childcare workers believed that snack-time should be a social moment spent together at the table. The fruit was chopped and divided, so that all children got some variety in the pieces of fruit on their plate. Usually, one plate was provided for two children. Childcare workers explained that this was a way to teach the children to share, but it could also help in letting children be role models to each other. The childcare workers also modelled behaviour themselves by using this moment to eat a piece of fruit with the children and trying to encourage them to try new food products. However, some less favourable practices were also described by the childcare workers. They were mostly related to pressuring children to finish their plate or take another bite, *‘we provide all children with a basic bowl* [with fruit] *which they are supposed to finish’ (CW5.1)*. Some childcare workers also mentioned that they used food as a reward: ‘*So, we say, “If you eat your fruit, then you get your cookie.” In this way we stimulate them to eat some fruit’ (CW3.2).* Influences in the social environment that were mentioned were predominantly parents and sometimes the community health service. *‘Then the community health service remarked that too much lemonade was served, and then we started thinking about why we actually do that’ (CM4).*


An important belief for childcare managers and workers to facilitate physical activity was providing a moment for the children to release their energy. They found it very important that the children go outside at least once a day. However, it was unclear whether this also happened in practice. It was mentioned that going outside was often skipped when there were time constraints. *‘You have to do certain activities in a certain planning and these often take up some time, making that you are not able to also go outside with the whole bunch’ (CM3)*. Other barriers to providing opportunities for physical activity were mentioned by the childcare workers. Some facilities had to share their outdoor playground with a primary school, and older children would be present on the playground at the same time as the young children, resulting in perceived unsafe situations. Parents did not always take into account that the childcare workers want to go outside with the children, despite various weather conditions. Therefore, children were not always suitably dressed. *‘Parents also find it too cold too soon … “Did you go outside to play?!” is what they say, and not all parents are always happy about that’ (CW2.1).* Both childcare managers and childcare staff expressed the need for an increased awareness of healthy nutrition and physical activity in the home setting. They believed that this would help them in stimulating healthy nutrition and physical activity in the childcare setting. *‘Teach parents what is healthy. There is a large group that thinks they know what is healthy, but maybe they can learn more, so that the child also better knows what is healthy’ (CM1)*. 

#### 3.3.3. Economic Environment

The economic childcare environment was primarily discussed with the childcare managers and workers. Almost all managers mentioned that they did not work with a pre-specified budget in regard to nutrition or physical activity. For example, if there was a need for new play materials, they evaluated whether this fit in the budget and then purchased it. *‘If I think it is worth the money, then it may cost something, and I do not really care that much about the costs’ (CM4)*. Some managers mentioned that financial cuts in childcare were something that influenced how much they could spend on healthy nutrition and physical activity. Many childcare workers at the pre-schools experienced a great monetary barrier regarding increasing the variety in fruits or vegetables or getting new play materials. They did not expect their organization to be able to provide fruit and vegetables for financial reasons. Furthermore, they felt that they could not expect parents to bring more unusual fruits due to the costs. *‘You cannot force parents to bring a pineapple if that puts someone to great expense’ (CW6.1)*. As fixed play materials often have to comply with strict safety regulations, they are too expensive to purchase, and thus childcare workers often did not ask for them. *‘Often there is no money to purchase new materials such as a slide, because they have to comply with all those safety rules. So, we cannot just go to IKEA to buy things that are cheaper’ (CW5.1)*. On the other hand, it was stated that budgets were sufficient to purchase small materials that can stimulate physical activity such as chalk pieces and bottles for blowing bubbles. 

#### 3.3.4. Political Environment

There was some variation between the locations regarding whether a nutrition-related policy was available. In centre-based childcare, food is provided by the organisation. Therefore, there was an elaborate policy around food products and drinks that are available, food safety, and permitted nutritional supervising practices. For pre-schools, where parents bring the majority of the food products for snack-time, the only policy was that the food products were supposed to be fruits.

The majority of all childcare locations had a policy aimed at stimulating a healthy treat for birthday celebrations. There was great variance in the adherence by parents to this policy and its implementation by childcare workers. This was partly due to the influence of the parents’ beliefs (perceived or real), and partly because the policy was formulated ambiguously and was not enforced, which makes it more a guideline. Some childcare workers expressed the need for a clear, strongly worded policy around birthday treats. They expected that this would help them in communicating it to the parents. *‘The policy does not prohibit sweets as a treat, it is only advised* [to provide something healthy]. *So, you cannot make parents accountable’ (CW9.3)*.

In general, no formal policy was formulated around physical activity. In particular, managers said that providing enough opportunities for physical activity is common practice for childcare workers and therefore does not need to be written down in formal policy. *‘At childcare we work a lot according to policies, protocols and rules and then I think: is it necessary for physical activity? We think some things do not need to be put down in policy and are part of the professionalism of our childcare workers’ (CM1)*. The rules and regulations imposed by the community health services were perceived as an important barrier to physical activity. *‘Well, a lot of things are bound to norms, a lot more than in educational institutions, and that limits the children’s physical activity. And we actually went over the top: watch out, look out, don’t do this, don’t do that, and that’s actually a very wrong development, just because the community health service tests us on it’(CM6).*


### 3.4. Cross-setting Influence of Childcare and Home Setting

The childcare managers and workers first talked about the responsibility of the parents to create a healthy home environment for their children. *‘So many things are being made our responsibility, but they come here only two or four mornings, yeah I think some things have to be for the parents. I do not think we have to do everything’ (CW5.1).* One childcare manager said that to promote a healthy lifestyle in young children, you need to involve the parents. *‘If you only do things here* [at childcare], *of course you achieve something, but not everything so I think that is very important* [to involve parents in healthy lifestyle changes]*’(CM7)*. There were several ways in which an influence of the home setting was experienced. Many comments related to dietary habits in the home environment. For example, children were not used to eating at the table, arrived at childcare without having had breakfast, or lacked skills to bite pieces of fruit due to still being bottle-fed. Childcare managers explained that they got requests from parents for special treatment of their child with regard to nutrition. Interestingly, this was often related to healthier nutritional choices, such as parents not wanting their children to drink sugar-sweetened beverages or eat sweet bread toppings. *‘There is a group of children of which the parents say, “They cannot have milk, they really cannot have sweets, they do not participate in birthday treats, they drink just water,” (CM5).* Mostly, the childcare managers did not go along with such individual requests, and stuck to their own nutrition policies. 

With regard to physical activity, a common remark was that children often arrive at childcare with unsuitable clothing for playing outside. *‘We really have to promote that children wear a jacket when it is cold weather, many do not bring a coat’ (CW8.1).* Some childcare workers mentioned that children at their facility lacked the locomotor skills to be able to join in all activities. They explained this as due to a less challenging home environment with regard to physical activity (e.g., lack of availability of certain play materials or space). 

Although it was not the focus of this paper, the interviews with the parents also revealed an influence of the childcare setting on the home setting. Parents mentioned that they noticed that their child ate more fruit and a greater variety of fruit, due to the fact that it is encouraged at childcare. *‘First at home he would not eat grapes, but now he comes here* [at the pre-school] *and he says ‘yum’ when he sees grapes’ (P15)*. However, it was also mentioned that going to childcare increased the intake of sugar-sweetened beverages. *‘They get lemonade too at my home, although I limit it to one glass a day … so I actually think then you have already got that here* [at the pre-school]. *But then I give him one extra at home’ (P25).* Some parents stated that they would rather see their child not drinking any sugar-sweetened beverages but took for granted what was served at the childcare facility.

### 3.5. Moderators

The interviewees mentioned several moderating factors, as described by the EnRG framework. Predominantly, demographic factors, personality factors, habit strength and awareness were discussed in the interviews. Ethnicity or cultural background was often mentioned. *‘You see that children with a different cultural background, they prefer something sweet instead of fruit, they are not too crazy about it, no’ (CW6.2).* Age was also mentioned, more often in relation to physical activity. Childcare workers experienced that younger children were more hesitant in joining in activities. *‘If those little ones go outside, the size* [of the playground] *is already overwhelming, then you have the older children running around, they just do not get to playing’ (CW6.1)*. Socio-economic status was mentioned as a moderator, such as the opportunities children have at home to be physically active and develop motor skills or the availability of healthy food products. Regarding personality, some child characteristics were mentioned such as allergies and preferences. They were mostly only related to nutrition. *‘Sometimes with little children it is quite difficult, because they are fussy with vegetables, for example’ (CM7)*. In relation to birthday treats, habit strength was mentioned. Childcare workers often thought that children, but even more so their parents, were used to a certain routine and would therefore want to stick to it. They mentioned that getting used to a new routine was important in accepting change. *‘Now they are used to it. They all know, the children too: “We do not have to bring anything. It is my birthday, and I can treat with those cookies* [provided by the childcare facility]*”’ (CW1.1).* The last moderating factor that appeared during the interviews was awareness. Firstly, this was generally seen in the lack of need for change, while some factors that were described can be considered unhealthy. Secondly, some participants described that being aware of, for example, the content of certain food products helped them in making healthier decisions. 

## 4. Discussion

This study aimed to explore facilitators and barriers to healthy nutrition and physical activity in childcare from different perspectives (childcare managers, childcare workers and parents). The EnRG framework was used to identify intrapersonal and environmental factors influencing childcare. All respondents expressed a positive attitude towards healthy nutrition and physical activity in childcare. However, less healthy aspects not always required attention in the respondents’ opinion. For example, the serving of sugar-sweetened beverages at the childcare location was approved or tolerated by all respondents. In the Netherlands, over half of the children consume more than two sugar-sweetened beverages per day [42], and schools can be an important venue for reducing sugar-sweetened beverage intake [43]. The majority of the respondents thought that being active is in the nature of young children and thus they need little encouragement to be sufficiently physically active. This perception may well be inaccurate, because research has shown that young children often do not meet physical activity guidelines, in particular for sedentariness [44,45,46]. Actively promoting physical activity by increasing awareness about low activity levels can be an important factor in decreasing the sedentariness of young children. 

Factors mentioned regarding the physical environment were most often related to the availability and variety of healthy food products and the availability of play materials and indoor and outdoor play space. This is comparable to other research that explored factors influencing nutrition and physical activity at the childcare centre [29,47,48,49]. Providing more, particularly portable, play materials has been part of intervention studies and had a positive effect on children’s moderate-to-vigorous physical activity (MVPA) [50,51]. A need for more natural elements in the playground was expressed to enhance its appeal to be more physically active. Previous research has shown the positive effects of natural elements on children’s physical activity, making this an important consideration for childcare interventions [52,53,54]. Exposure to new food products has been shown to be an effective strategy in helping children eat and like these products [55]. Changing the physical environment through providing healthy food products and play materials could be an effective measure to overcome the perceived barriers in childcare. However, there was an apparent difference in the influences perceived in the physical environment between the childcare managers and parents on the one hand and the childcare workers on the other. The childcare managers and parents were more positive and described the opportunities that the physical environment offered for healthy EBRB, while the childcare workers were more negative and described barriers they felt needed to be overcome for healthy EBRB. One explanation could be that the managers worked at different childcare facilities where indeed the physical environment was more facilitating for healthy EBRB. Another explanation could be that there is a discrepancy in the expected role of the childcare worker regarding healthy EBRB. Studies have described that for physical activity childcare workers often see their role as supervising and guarding safety, but not actively participating in activities and that this may differ from the expected role from childcare managers [56,57]. As a result, childcare workers might tend to attribute influences on children’s behaviour more externally, in this case the physical environment, while a childcare manager might rely more on the childcare worker’s behaviour (i.e., sociocultural environment) in relation to the opportunities in the physical environment. As one is not more important than the other, this underlines the combined influence of different types of environments as suggested by socio-ecological models. Therefore, this is something important to take into account in intervention development and implementation. A last explanation could be that this difference exists because the childcare managers are too far distanced from daily practice and, therefore, cannot accurately estimate the influence of the physical environment. In our sample, though, more than half of the managers also had experience as a childcare worker and thus may be able to understand the influence of the physical environment in daily practice. 

The sociocultural environment is predominantly formed by the behaviour of the childcare workers. Some favourable practices were described by the childcare workers (e.g., modelling of healthy eating), but also some unfavourable ones (e.g., using food as a reward) [28]. It appeared from the interviews that almost all participants agreed with how things are run at the childcare centre. This implies that childcare workers are not sufficiently aware of the practices they can use to influence the behaviour of the children, which is also supported by previous research [58]. This may be an important aspect to pay attention to in intervention development, for example when training childcare workers. Furthermore, some other influences on the sociocultural environment were mentioned. Parents were perceived as an important influence on the ability to promote healthy nutrition and physical activity in childcare. For example, through the clothes they let their children wear and healthy eating habits they teach their children at home. Aspects in the physical environment also influenced the behaviour of the childcare workers (e.g., available time, scheduling problems with the primary school), which indirectly influenced the sociocultural environment. 

Factors in the economic environment mostly concerned financial means. The economic environment has not been extensively studied before in the childcare setting yet. A study of family childcare did mention financial considerations as an influence on food choices [31,32]. The high cost of healthy food is something that is often mentioned by parents, especially ones with a low socio-economic background [48]. This is in line with the concern expressed by childcare workers in the current study that it would burden the parents of the children with high costs to bring more unusual fruits. There was a difference noted between the childcare workers and managers regarding perceived economic factors for physical activity. Many childcare workers expressed a barrier to purchasing new play equipment due to high costs. Childcare managers expressed that they found it more important to know whether new equipment was needed than the cost of this equipment. It could be that childcare managers gave more social-desirable answers and in reality are more cautious in purchasing new equipment. Further, this might again point towards a combined influence of different types of environments (the economic and physical environment). In intervention development, this may mean that both environments should be taken into account in order to assure intervention effectiveness. 

In the political environment, some distinct differences were seen. The different types of facilities (pre-school or centre-based day care) influenced whether an elaborate nutrition policy was in place. This was mostly related to whether food was provided by the facility or not. The lack of an institutional policy led to perceived ambiguity by the childcare workers, particularly relating to birthday treats. Most childcare workers felt that birthday treats needed to change, but they did not feel supported by their institutional policy. The Netherlands Nutrition Centre provides an example policy statement that many institutions use to formulate their own policies. The specification and translation of this example into institutional policy appear particularly important. Previous research confirms the positive influence that policies could have on EBRB of children in childcare [59]. It is important to note that even with policies in place, the translation of policies into childcare staff’s practices is still very important for the promotion of healthy EBRB in young children [58,59]. Another difference was seen between nutrition and physical activity. While a nutrition policy was often mentioned, this was not the case for physical activity. Providing sufficient opportunities for physical activity (e.g., by going outside) was mostly assumed to be something that is just done and does not require a formal policy. This lack of a formal PA policy, which is also seen in other studies, is explained by the common, mistaken notion that young children are inherently active [60]. Policies and guidelines have the potential to support physical activity through increasing the quality of play times; for example, by setting structured play times [61]; ensuring appropriate clothing [62]; or childcare staff behaviours during play time [63]. It is important to take into account the often mentioned remark in this study not to overload the staff with regulations and policies. This could have adverse effects, as evident from the regulations of the community health services that are limiting instead of promoting physical activity. 

Weather as a perceived barrier to physical activity is often described in the literature [47,64,65]. Interestingly, in this study, weather was not always seen as a barrier by the childcare workers and managers. The Dutch climate is quite moderate, with few extremes compared to other countries. This may explain the limited perceived influence of the weather itself. Some care should be taken with this finding because it is not clear whether childcare workers really went outside in all types of weather. Some childcare workers did express weather as a barrier, in particular hot and sunny weather in combination with lack of shade in the playground, which is also in line with previous research [52,64]. Providing shade in order to promote physical activity is something that is often overlooked and not part of intervention research. This may be a factor to take into account in future research as it may eliminate a perceived barrier to going outside. The childcare workers did experience a barrier if the clothing of the children was not suitable for going outside. This is also something seen in other studies [30,62].

The participants in the interviews talked about moderating factors that could influence the effect of the facilitators and barriers for children’s EBRB. They were mostly in line with existing knowledge on moderating factors such as the child’s age, socio-economic status, cultural background and characteristics [66,67,68,69]. An interesting finding involved the possible influence of awareness. Several unfavourable environmental factors were described for which a need for change was not always expressed. This could be greatly influenced by a lack of awareness, for example, in relation to the use of unfavourable practices by childcare staff. For intervention development, it is important to take this into account and focus not only on ‘how’ to promote healthy children’s EBRB, but also on ‘why’. 

An important theme in the interviews was the influence of parents and the home setting on EBRBs of children in the childcare setting. Important findings in this study were that childcare workers and managers both stressed the importance of involving parents and the home setting in the promotion of healthy behaviours in children. They felt they had limited opportunities (e.g., the amount of time children spent at preschool in comparison to the time they spent at home) to influence the lifestyle of young children. They also expressed perceived barriers in the childcare setting regarding nutrition behaviour and physical activity through the parents’ practices or environmental factors in the home setting. These influences of the home setting might also elicit childcare workers to knowingly use unfavourable practices for fear of parental reactions [70]. On the other hand, the interviews with parents revealed both positive and negative influences of the childcare setting on the home setting. These findings indicate an interaction between the childcare and home settings, which is hypothesised in the ecological systems perspective as the mesosystem [71,72,73]. Such a mesosystem acknowledges not only the interaction between personal characteristics and environmental influences, but also the interaction between different types of environments and between environmental settings, like the childcare and home settings [73]. The interaction between determinants is often underrepresented in the current literature on children’s EBRB [73]. Other qualitative studies have described this interaction between home and childcare settings [29,48,74], and a recent quantitative study was the first to show the existence of this mesosystem and its influence on child outcomes [75]. Recent reviews on the effectiveness of interventions do highlight the importance of this interaction by recommending a comprehensive, multi-component approach (i.e., combining home and childcare settings) in prevention interventions for childhood overweight and obesity [76,77]. 

This study has some strengths and limitations that should be taken into account when interpreting its results. One limitation is the possibility of social desirability in the participants’ answers. Due to the nature of the study, it is possible that the participants did not describe the situation as it was, but as they would want it to be or think it should be. The parents in this study might not want to criticise the childcare staff at their facility due to their personal relationship. They may have been overly positive. Another limitation is that the interviews with the parents were quite short. Therefore, it was not always possible to explore in depth their views on nutrition and physical activity at the childcare centre. The economic and political aspects were not discussed with the parents. This may have resulted in an underrepresentation of the parents’ opinion in this study. However, combining different stakeholders in this study enabled us to explore all types of environments from different viewpoints. The influential factors in the types of environment were also explored with those experiencing them directly. A last limitation is related to the generalizability of the results. Due to the qualitative study design and the recruitment methods, the results of this study may not be generalizable to other populations. Differences in childcare systems, nutritional and physical activity (cultural) habits, and local and national policies may influence the generalizability of these results. However, similar influential factors may be applicable in other regions as several factors found in this study overlap with previous research into determinants and facilitators and barriers to children’s EBRB in childcare (e.g., [30,48,56]). With this study, our qualitative knowledge on influential factors has increased on the influential factors in different types of environments. Quantitative studies are needed to evaluate the robustness of these results, in particular on the existence of an interaction between these environments. Nonetheless, as formative research for intervention development, it is important to explore influential factors specifically in the context in which the intervention will be implemented [78]. Adaptation of interventions to their context can be pivotal in their implementation and sustainability [79]. 

## 5. Conclusions

The current study gave us some insight in the obesogenity of the childcare and home environment. Several facilitating and hindering factors were identified in all types of environments. The promotion of healthy EBRB in young children in childcare is something that is considered important by the different stakeholders. Although a need for change was not always expressed, opportunities for improvements in childcare to promote healthy EBRB in young children were revealed. An interaction between the childcare and home settings was recognised. Therefore, a mesosystem approach seems necessary in intervention development in which intervention components are aligned in both the childcare and the home setting. 

## Figures and Tables

**Table 1 ijerph-15-02909-t001:** Demographics of childcare managers and childcare workers.

	Childcare Managers (N = 9)	Childcare Workers (N = 23)
Mean age in years (range)	40.1 (29–69)	49.8 (19–65)
Gender Female (%) Male (%)	9 (100) n.a.	23 (100) n.a.
Educational level ^a^		
Lower vocational pedagogical education	0	7 (30.4)
Lower vocational social work	2 (22.2)	2 (8.7)
Higher vocational pedagogical education	2 (22.2)	9 (39.1)
Higher vocational social work	0	1 (4.3)
Other	4 (44.4) ^b^	4 (17.4) ^c^
Average working years (SD)	6.3 (6.1)	17.6 (8.4)
Previous or current experience as childcare worker (%)	5 (55.5)	n.a.

^a^ Education of one manager was unknown; ^b^ All four childcare managers had a higher vocational education, but not pedagogical; ^c^ One childcare worker had a higher vocational education not pedagogical, two had a lower vocational education not pedagogical and one was still in training.

**Table 2 ijerph-15-02909-t002:** Demographics of parents.

	Parents (N = 31 ^a^)
Mean age in years (range)	31.7 (22–41)
Gender ^b^	
Female (%)	26 (81.3
Male (%)	7 (18.7)
Educational level ^c^	
Low (%)	8 (25.0)
Medium (%)	11 (34.4)
High (%)	12 (37.5)
Employment status	
Unemployed (%)	16 (51.6)
Employed^d^ (%)	15 (48.4)
Full-time (%)	10 (66.7)
Part-time (%)	4 (26.7)
Relational status	
In a relationship (%)	27 (87.1)
Not in a relationship (%)	4 (12.9)
Average number of children (range)	1.8 (1–4)

^a^ Characteristics of two parents were unavailable; ^b^ Gender was based on tone of voice, and therefore available for all participating parents; ^c^ Based on ISCED-97 classification: low equals levels 0, 1 and 2; medium equals levels 3 and 4; and high equals levels 5 and 6 [41]; ^d^ Working hours of one parent were unknown.

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
