# Peer review of "Healthy Nutrition and Physical Activity in Childcare: Views from Childcare Managers, Childcare Workers and Parents on Influential Factors"

_ijerph, 2018, doi:10.3390/ijerph15122909_

Reviewer 1 Report

Thank you for the opportunity to review ‘Healthy nutrition and physical activity in childcare:

views from childcare managers, childcare workers and parents on influential factors. This is an important paper using qualitative methods to explore the views of childcare managers, workers and parents on facilitators and barriers to healthy EBRB’s in childcare environments.

The following comments have been provided to assist with improving the quality of this paper in order to be published in IJERPH.

Abstract:

Line 16 – remove capital on ‘Gain’.

Line 20 – do you mean improvement of healthy EBRB’s in childcare?

Key words: Add ‘behaviour’

Introduction:

Line 2 - please provide some additional primary evidence for the statement about children now growing up in an obesogenic environment, specifically for the high energy dense foods and low levels of physical activity / sedentary behaviours.

Line 36 – Do you mean obesity-related diseases?  If so, please state this explicitly.

Line 36 – Please state who has increased their attention on prevention of childhood overweight and obesity. 

Line 50 – Please identify barriers and facilitators to what? In the childcare setting.

Lines 70 – 72 – Please explicitly state your aims at the end of the introduction.

Materials and Methods:

Line 103 – Please indicate the ethical approval number and institution that approved the study.

Results:

Line 127 – change “lasted 34 (range: 21-60) minutes” to read: lasted 34 minutes (range: 21-60)

Table 1 – Please add +- Standard Deviation (SD) to Working years.

Section 3.3.4 – I wonder if this heading could be titled: ? Policies, guidelines and regulations?  The term ‘political environment’ sounds more like it is related to government laws and acts.  Maybe this is part of the framework though so may not be able to be changed.  If it can be changed, I would consider re-titling this section, so it is clearer for the reader what it is about.

Line 298 – Please provide a quote as evidence for the statement: Childcare managers explained that they often got requests from parents for special treatment of their child with regard to nutrition. Interestingly, this was often related to healthier nutritional choices, such as parents not wanting their children to drink sugar-sweetened beverages or eat sweet bread toppings.

Discussion:

Line 408 – Again the term ‘political environment’ does not seem to accurately reflect the points being discussed.  If you keep this term, it will need some further explanations in the methods as to what it includes.

Line 475 – regarding generalizability of the results to other populations.  I do not see any demographic data that suggests the participants of this study represent the wider area (either within the day care setting or outside it in Southern Netherlands).  The authors should amend the paragraph so as to not lead the reader to believe that the results of this study may be generalizable to Southern Netherlands either).

Overall – This is a very interesting study that reads well and will be a valuable addition to the mostly quantitative research undertaken in these environments regarding contributors to child obesity.  It would be appropriate for the authors to link their findings back to their original comments regarding obesogenic environments.  I look forward to seeing this paper published if the above-mentioned comments can be addressed.

Author Response

Reviewer 1

Thank you for the opportunity to review ‘Healthy nutrition and physical activity in childcare:

views from childcare managers, childcare workers and parents on influential factors. This is an important paper using qualitative methods to explore the views of childcare managers, workers and parents on facilitators and barriers to healthy EBRB’s in childcare environments.

The following comments have been provided to assist with improving the quality of this paper in order to be published in IJERPH.

Abstract:

Line 16 – remove capital on ‘Gain’.

Response: This was written with a capital because it is part of the name of the EnRG framework that is used in this study. To clarify this, we have written the name between quotation marks and added the abbreviation.

Correction:

                Line 16-17: Content analysis was performed using the ‘Environmental Research framework for weight Gain prevention’ (EnRG framework) …

Line 20 – do you mean improvement of healthy EBRB’s in childcare?

Response: Indeed, we mean improvements for healthy EBRB’s. We have added this to the sentence.

Correction:

                Line 20-21: Although a need for change was not always expressed, the interviews revealed opportunities for improvement of healthy EBRB in childcare.

Key words: Add ‘behaviour’

Response: Thank you for your suggestion, we have added behaviour as a keyword. In order to keep to the authors’ guidelines, we have removed ‘facilitators’.

Introduction:

Line 2 - please provide some additional primary evidence for the statement about children now growing up in an obesogenic environment, specifically for the high energy dense foods and low levels of physical activity / sedentary behaviours.

Response: We have provided additional primary evidence.

Line 36 – Do you mean obesity-related diseases?  If so, please state this explicitly.

Response: Yes, indeed we meant overweight and obesity-related diseases, so we made it more explicit.

Correction:

                Line 42-43: Furthermore, weight status between two and six years of age is most predictive for adult weight status and overweight and obesity-related diseases [10, 11].

Line 36 – Please state who has increased their attention on prevention of childhood overweight and obesity.

Response: We have added ‘In practice and research’ to this line. The increased attention can be seen in childcare and preschools as they are implementing more and more programs that are aimed at changing children’s EBRB. This is reflected in the references provided which are related to the research that is performed on these programs.

Correction:

                Line 43:  In practice and research increased attention has been paid by researchers to the prevention of childhood overweight and obesity …

Line 50 – Please identify barriers and facilitators to what? In the childcare setting.

Response: We have added ‘of healthy EBRB’ in this sentence to clarify to what the barriers and facilitators are related.

Correction:

                Line 56-57: … a number of studies have examined perceived facilitators and barriers of healthy EBRB in the childcare setting.

Lines 70 – 72 – Please explicitly state your aims at the end of the introduction.

Response: We have rephrased our last sentence of the introduction to clearly state the aims of the study while incorporating the inclusion of different stakeholders, which was previously formulated in a separate sentence.

Correction:

                Line 75-78: The aim of this study was to explore facilitators and barriers in childcare to promote healthy nutrition and physical activity, using the EnRG framework and taking the views of different stakeholders (i.e. childcare managers, childcare workers and parents) into account.

Materials and Methods:

Line 103 – Please indicate the ethical approval number and institution that approved the study.

Response: Our apologies for not providing the ethical statement for this study in the first version of the manuscript. This study was performed as part of a larger research project which was approved by the Maastricht University Medical Centre+ Medical Ethics Committee (METC163022). We have added this statement to the manuscript in the section ‘study sample and recruitment’.

Corrections:

                Line 109-111: The Maastricht University Medical Centre+ Medical Ethics Committee reviewed and approved this study as part of a larger research project (METC163022). 

Results:

Line 127 – change “lasted 34 (range: 21-60) minutes” to read: lasted 34 minutes (range: 21-60)

Response: We have changed the sentence.

Correction:

                Line 138: lasted 34 minutes (range: 21-60) on average with childcare workers

Table 1 – Please add +- Standard Deviation (SD) to Working years.

Response: We have added the Standard Deviation in the table.

Correction:

                Table 1:

Average working years (SD)

6.3 (6.1)

17.6 (8.4)

Section 3.3.4 – I wonder if this heading could be titled: ? Policies, guidelines and regulations?  The term ‘political environment’ sounds more like it is related to government laws and acts.  Maybe this is part of the framework though so may not be able to be changed.  If it can be changed, I would consider re-titling this section, so it is clearer for the reader what it is about.

Response: We understand that political environment may induce confusion. Within the framework the political environment is defined as the laws, regulations, policies and institutional rules that influence nutrition and physical activity, based on the ANGELO framework [1]. This can be applicable for both micro settings, such as the childcare or home setting, and macro settings such as society. As we are looking specifically into the childcare micro setting, the political environment predominantly reflects the policies, rules and regulations that are encountered in this micro setting as they arose from the interviews. As political environment is part of the terminology used in the ANGELO framework we chose not to change the heading of this section, but we added more information on what we consider to be the political environment in this study in the methods section.

1.            Swinburn, B.; Egger, G.; Raza, F., Dissecting obesogenic environments: the development and application of a framework for identifying and prioritizing environmental interventions for obesity. Prev. Med. 1999, 29, (6 Pt 1), 563-70. 10.1006/pmed.1999.0585

Correction:

                Line 132-134: For the construct ‘political environment’ content describing rules, regulations, policies regarding nutrition or physical activity in the childcare setting are considered shaping the political environment.

Line 298 – Please provide a quote as evidence for the statement: Childcare managers explained that they often got requests from parents for special treatment of their child with regard to nutrition. Interestingly, this was often related to healthier nutritional choices, such as parents not wanting their children to drink sugar-sweetened beverages or eat sweet bread toppings.

Response: We have added a quote, which is also provided in the Supplementary material. Further, we changed ‘these’ into ‘such’ in the following sentence, as we saw now that this would be more appropriate regarding the reported result.

Correction:

                Line 312-314: ‘There is a group of children of which the parents say, “They cannot have milk, they really cannot have sweets, they do not participate in birthday treats, they drink just water,” (CM5).

Discussion:

Line 408 – Again the term ‘political environment’ does not seem to accurately reflect the points being discussed.  If you keep this term, it will need some further explanations in the methods as to what it includes.

Response: This comment relates to the previous comment on ‘political environment’. We have clarified what we considered to be the political environment in the methods section.

Line 475 – regarding generalizability of the results to other populations.  I do not see any demographic data that suggests the participants of this study represent the wider area (either within the day care setting or outside it in Southern Netherlands).  The authors should amend the paragraph so as to not lead the reader to believe that the results of this study may be generalizable to Southern Netherlands either).

Response: We agree with your comment that the results of this study may not be generalizable for other populations, even in Southern Netherlands. Therefore, we have adjusted this section of the manuscript to reflect better on the generalizability of the results and the value of these results.

Corrections:

                Line 489-490: Due to the qualitative study design and the recruitment methods, the results of this study may not be generalizable to other populations, although similar influential factors may be applicable in other regions.

Overall – This is a very interesting study that reads well and will be a valuable addition to the mostly quantitative research undertaken in these environments regarding contributors to child obesity.  It would be appropriate for the authors to link their findings back to their original comments regarding obesogenic environments.  I look forward to seeing this paper published if the above-mentioned comments can be addressed.

Response: Thank you for your compliments and your suggestions for improvement. We have made a link with obesogenic environments in our conclusions.

Corrections:

                Line 512-513: The current study gave us some insight in the obesogenity of the childcare and home environment. Several facilitating and hindering factors were identified in all types of environments.

Reviewer 2 Report

Comments

Comments and Suggestions for Authors: 

Abstract

1.Very clear summary of the findings of the article. Please add brief definitions of energy balance-related behaviors. 

2.Please add Netherlands in the Keywords.

Introduction

The author does a good job describing EREB and related factors, and clearly state the purpose of the present study. A bit more background information about childhood overweight and obesity in young children in the Netherlands would be useful.  How do these compare with other countries?

Materials and Methods

Very clear description.

1.How long did an interview take?

2. It would be better described more about Data coding and analysis. 

Results

The results section is too long, please reduce and avoid repetition. 

Discussion

1.For Line 344-345 “For example, the serving of sugar-sweetened beverages at the childcare location was approved or tolerated by all respondents”, it will be better that discuss this with the SSB consumption among young children in Netherland or around the world.

2.For the political environment, the authors discussed “lack of an institutional policy”, how about the national policy?

Author Response

Abstract

1.Very clear summary of the findings of the article. Please add brief definitions of energy balance-related behaviors.

Response 1: Thank you for your suggestion and we have added some examples of energy balance-related behaviours in the abstract.

Correction:

line 13: The childcare setting plays an important role in children’s energy balance-related behaviours (EBRB), such as physical activity, sedentary behaviour and healthy nutrition. 

2.Please add Netherlands in the Keywords.

Response 2: Thank you for your suggestion. We have added Netherlands as a keyword. In order to comply with the authors’ guidelines, we have removed ‘barriers’ from the list of keywords.

Introduction

The author does a good job describing EREB and related factors, and clearly state the purpose of the present study. A bit more background information about childhood overweight and obesity in young children in the Netherlands would be useful.  How do these compare with other countries?

Response:  Thank you for the compliment and the suggestion for improvement. We have added information on the prevalence of childhood overweight and obesity in the Netherlands, specifically for the age-group that is the topic of this paper. Further, we have added some information on the comparison with other countries.

Corrections:

Line 34-40: In the Netherlands, 8.0% of 2-year old boys and 8.3% of 2-year old girls are overweight including 0.7% obese in both boys and girls and these numbers increase to 9.1% (boys, overweight), 16.3 % (girls, overweight), 1.1% (boys, obese) and 2.6% (girls, obese) for 4-year old children [5]. These numbers are comparable to the prevalence of overweight and obesity in other Northern European countries, but are fairly favourable compared to the prevalence in other Western countries [7, 8].  As the prevalence of childhood overweight and obesity is expected to keep rising, the prevention of childhood overweight and obesity is still an important public health issue [8].

Materials and Methods

Very clear description.

1.How long did an interview take?

Response 1: We described the duration of the interviews at the beginning of the results section. These were as follows: The interviews with childcare managers lasted on average 42 minutes (range: 32-50), lasted 34 minutes (range: 21-60) on average with childcare workers, and almost 10 minutes (range: 4-23) on average with parents.

2. It would be better described more about Data coding and analysis.

Response 2: We have added information on the codes used during data analysis in the paragraph of data processing and analysis.

Correction:

Line 124 – 127: The constructs of this framework (e.g. the types of environments and cognitive determinants) formed the basis of the content analysis. Additionally, codes were used to increase specificity, such as ‘nutrition or physical activity’, ‘indoor- or outdoor- play area’, or ‘influence of other preschools or other preschool teachers’ that arose from the data.

Results

The results section is too long, please reduce and avoid repetition.

Response:  We are aware of the length of the results section, but feel that this is in part unavoidable because we are reporting qualitative results. Further, we report on both nutrition and physical activity and we try to reflect the views of the multiple stakeholders in the most appropriate way. Nonetheless, we have critically revised the results section and removed some repetition based on your remark.

Discussion

1.For Line 344-345 “For example, the serving of sugar-sweetened beverages at the childcare location was approved or tolerated by all respondents”, it will be better that discuss this with the SSB consumption among young children in Netherland or around the world.

Response 1: We have added a sentence on SSB consumption in children and the effects of high SSB consumption.

Correction:

                Line 361-363: In the Netherlands, over half of the children consume more than two sugar-sweetened beverages per day [42], and schools can be an important venue for reducing sugar-sweetened beverage intake [43].   

2.For the political environment, the authors discussed “lack of an institutional policy”, how about the national policy?

Response 2: The Netherlands Nutrition Centre provides an example policy statement for nutrition in childcare. Many organisations use this example to formulate their own policies. Although, this example urges to promote healthy birthday treats, formulation is still very ambiguous. Therefore, the specification and translation of this example into institutional policy is particularly important. It appeared in our study that the childcare workers felt that this translation could be more strong-worded as to help them use the policy in practice. As this was emphasized by our participants we focused on the institutional policy in our discussion. We have added some information on the national policy in our discussion.

Corrections:

                Line 434 – 436: The Netherlands Nutrition Centre provides an example policy statement that many institutions use to formulate their own policies. The specification and translation of this example into institutional policy appears particularly important.    

Reviewer 3 Report

 Thank you for the opportunity to review “Healthy nutrition and physical activity in childcare: views from childcare managers, childcare workers and parents on influential factors” for Int. J. Environ. Res. Public Health. This interesting and important paper analyzes facilitators and barriers of healthy children’s energy balance-related behaviors (EBRB) in childcare in a comprehensive way, from the perspective of three crucial stakeholders: childcare managers, childcare workers, and parents. Overall, the paper is very well-written. But there are some minor comments below for the authors to consider:

1. I was expecting to see some statistics on the prevalence of being overweight or obese among children in the Netherlands (after line 38), and preferably by age group of children considered as the sample of interest in this study. But authors never do it. I would strongly encourage, authors provide such statistics and even better to compare those with maybe overall OECD countries, as to inform readers how acute the problem is in the Netherland.

2. Between lone 79 and 83, there is a lot of specific information provided, hence it would be appropriate to provide references, even if one was provided before.

3. Please provide a reference for the sentence starting in line 85 and ending in line 86.

4. In the 2.2. “Study sample and recruitment section” would be appropriate to provide a brief description of the responsibilities of the Childcare managers and childcare workers and education hey should have and appropriate references.

5. Authors do not provide whether an appropriate IRB (Institutional Review Board) was obtained to conduct this research or whether it was exempt based on any grounds.

6. In Table 1, please provide the education specifics of other four childcare managers and maybe other four child care managers in a footnote.

7. In Table 2, please provide more specific information as to what low, medium, and high education mean. For example, less than high school, high school only, some college, or college, etc.

8. The 3.3.4. Political Environment seems to have a loaded connotation and might imply the influence of a larger political system, which is of course not the case in your study. I am not familiar with the framework the authors have used but wondered if this heading can be named as maybe Policy Environment referring to the institutional level laws, guidelines, and regulations. But this is, of course, optional and depends on the definition of the framework.

9. Somewhere between lines 68 and 72 please provide a brief description of different types of environment: sociocultural, physical, economic and political.

10. In the discussion about political environment section between lines 408 and 426, I wonder whether authors would consider suggesting to regulate nutrition-related policies, for example, to ban the use of sugar-sweetened beverages and encourage the use of fruits and vegetables in the centre-based childcare where the organisation provides food.

11. Finally, I have a concern with the statement about lack of generalizability in lines 476 and 477. Given the qualitative nature of the study and lack of random sampling, findings are not only generalizable to other regions of the Netherland, but I also assume for the South region where the study was conducted. I would suggest authors add a language that does not apply generalizability for the South region.

12. My another major concern is with the statement in line starting in line 480 “This suggests that similar factors influence children’s EBRB in western countries, and that makes this research also valuable for other regions.” This statement is contrary to the preceding comment that the findings are not generalizable. Authors imply now that all western countries share similar factors influencing children’s EBRB. I would suggest changing the language and maybe, as per the primary purpose of the qualitative study imply what authors have learned more in-depth about potential factors influencing EBRB so future research can examine whether those associations are robust or causal using representative cross-sectional or longitudinal data.

Author Response

Thank you for the opportunity to review “Healthy nutrition and physical activity in childcare: views from childcare managers, childcare workers and parents on influential factors” for Int. J. Environ. Res. Public Health. This interesting and important paper analyzes facilitators and barriers of healthy children’s energy balance-related behaviors (EBRB) in childcare in a comprehensive way, from the perspective of three crucial stakeholders: childcare managers, childcare workers, and parents. Overall, the paper is very well-written. But there are some minor comments below for the authors to consider:

1.       I was expecting to see some statistics on the prevalence of being overweight or obese among children in the Netherlands (after line 38), and preferably by age group of children considered as the sample of interest in this study. But authors never do it. I would strongly encourage, authors provide such statistics and even better to compare those with maybe overall OECD countries, as to inform readers how acute the problem is in the Netherland.

Response 1:  We have added information on the prevalence of childhood overweight and obesity in the Netherlands, specific for the age-group that is the topic of this paper. Further, we have added some information on the comparison with other countries.

Corrections:

Line 34-40: In the Netherlands, 8.0% of 2-year old boys and 8.3% of 2-year old girls are overweight including 0.7% obese in both boys and girls and these numbers increase to 9.1% (boys, overweight), 16.3 % (girls, overweight), 1.1% (boys, obese) and 2.6% (girls, obese) for 4-year old children [5]. These numbers are comparable to the prevalence of overweight and obesity in other Northern European countries, but are fairly favourable compared to the prevalence in other Western countries [7, 8].  As the prevalence of childhood overweight and obesity is expected to keep rising, the prevention of childhood overweight and obesity is still an important public health issue [8].

2. Between lone 79 and 83, there is a lot of specific information provided, hence it would be appropriate to provide references, even if one was provided before.

Response 2: We have added the references where they are appropriate. Some of the information in based on the knowledge of the authors on the childcare system in the Netherlands.

Correction:

                Line 85-91: The current study focuses on the first type: formal centre-based childcare, with a specific focus on pre-schoolers. Pre-schools provide half-day childcare with a focus on playful learning to prepare children for primary school. In this type of childcare, there is only one moment during which children consume food (snack time), and they often bring their own food. Children between 2 and 4 years old can attend preschool [37]. Centre-based childcare provides whole-day childcare and usually focuses less on educational goals [37]. In this type of childcare, there are several moments during which children consume food, and the childcare institutions mostly provide the food products. Children 0-4 years old are able to attend centre-based childcare [37].

3.       Please provide a reference for the sentence starting in line 85 and ending in line 86.

Response 3: We have provided a reference for this sentence.

Correction:

                Line 91-92: Parents can receive a general childcare benefit for formal childcare from the government, based on their working hours and income [38].

4. In the 2.2. “Study sample and recruitment section” would be appropriate to provide a brief description of the responsibilities of the Childcare managers and childcare workers and education hey should have and appropriate references.

Response 4:  We included extra information on the responsibilities of both the childcare managers and childcare workers in this section.

Correction:

                Line 97-100: All childcare managers had supervisory and policy-making responsibilities. Childcare workers were responsible for the daily supervision of the children and provision of the educational activities at preschool. Childcare workers should be minimally trained with a lower vocational pedagogical education [39].

5. Authors do not provide whether an appropriate IRB (Institutional Review Board) was obtained to conduct this research or whether it was exempt based on any grounds.

Response 5: Our apologies for not providing the ethical statement for this study in the first version of the manuscript. This study was performed as part of a larger research project which was approved by the Maastricht University Medical Centre+ Medical Ethics Committee (METC163022). We have added this statement to the manuscript in the section ‘study sample and recruitment’.

Corrections:

                Line 109-111: The Maastricht University Medical Centre+ Medical Ethics Committee reviewed and approved this study as part of a larger research project (METC163022). 

6. In Table 1, please provide the education specifics of other four childcare managers and maybe other four child care managers in a footnote.

Response 6: We have added the specifics in the footnote as you suggested.

Correction:

                Footnote table 1: bAll four childcare managers had a higher vocational education, but not pedagogical, cOne childcare worker had a higher vocational education not pedagogical, two had a lower vocational education not pedagogical and one was still in training

7. In Table 2, please provide more specific information as to what low, medium, and high education mean. For example, less than high school, high school only, some college, or college, etc.

Response 7: The qualification of education was based on the ISCED-97 classification to increase international comparability. The levels 0, 1 and 2 were considered low education (max lower secondary level of education); the levels 3 and 4 medium (max post-secondary, not tertiary education); and the levels 5 and 6 high (max second stage of tertiary education). We have included the reference to the ISCED-97 manual to justify the choices made. 

Correction:

                Footnote table 2: cBased on ISCED-97 classification: low equals levels 0,1 and 2; medium equals levels 3 and 4; and high equals levels 5 and 6 [41];

8. The 3.3.4. Political Environment seems to have a loaded connotation and might imply the influence of a larger political system, which is of course not the case in your study. I am not familiar with the framework the authors have used but wondered if this heading can be named as maybe Policy Environment referring to the institutional level laws, guidelines, and regulations. But this is, of course, optional and depends on the definition of the framework.

Response 8: We understand that political environment may induce confusion. Within the framework the political environment is defined as the laws, regulations, policies and institutional rules that influence nutrition and physical activity, based on the ANGELO framework. This can be applicable for both micro settings, such as the childcare or home setting, and macro settings such as society. As we are looking specifically into the childcare micro setting, the political environment predominantly reflects the policies, rules and regulations that are encountered in this micro setting as they arose from the interviews. As political environment is part of the terminology used in the ANGELO framework we chose not to change the heading of this section, but we added more information on what we consider to be the political environment in this study in the methods section.

Correction:

                Line 132-134: For the construct ‘political environment’ content describing rules, regulations, policies regarding nutrition or physical activity in the childcare setting are considered shaping the political environment.

9. Somewhere between lines 68 and 72 please provide a brief description of different types of environment: sociocultural, physical, economic and political.

Response 9: We have added a brief description of the different environmental types.

Correction:

                Line 74-76: The EnRG framework describes different types of environment, based on the ANGELO framework, namely sociocultural (what is the social and cultural background), physical (what is available), economic (what are the costs) and political (what are the rules) [35, 36].

10. In the discussion about political environment section between lines 408 and 426, I wonder whether authors would consider suggesting to regulate nutrition-related policies, for example, to ban the use of sugar-sweetened beverages and encourage the use of fruits and vegetables in the centre-based childcare where the organisation provides food.

Response 10: Policies could be an important aid in promoting healthy EBRB in children. It may be tempting to just say that we need stronger policies that ban certain unhealthy foods from, for example childcare and preschools. However, as we also describe in the discussion, policy alone is often not sufficient for behavioural change. In addition, what we also saw in our study is that the public opinion may not be ready for these types of rigorous policies as most respondents were satisfied with how nutrition was organized in the childcare setting. We believe that for now institutional policies around nutrition and physical activity may be more helpful, in particular if they are formulated with broad support from the organisation which may result in a better implementation of the policy.  

11. Finally, I have a concern with the statement about lack of generalizability in lines 476 and 477. Given the qualitative nature of the study and lack of random sampling, findings are not only generalizable to other regions of the Netherland, but I also assume for the South region where the study was conducted. I would suggest authors add a language that does not apply generalizability for the South region.

12. My another major concern is with the statement in line starting in line 480 “This suggests that similar factors influence children’s EBRB in western countries, and that makes this research also valuable for other regions.” This statement is contrary to the preceding comment that the findings are not generalizable. Authors imply now that all western countries share similar factors influencing children’s EBRB. I would suggest changing the language and maybe, as per the primary purpose of the qualitative study imply what authors have learned more in-depth about potential factors influencing EBRB so future research can examine whether those associations are robust or causal using representative cross-sectional or longitudinal data.

Response 11 & 12: We agree with your comment that the results of this study are indeed not generalizable for other populations, even in Southern Netherlands. Therefore, we have adjusted this section of the manuscript to reflect better on the generalizability of the results and the value of these results. In addition, we took into account the point you’ve raised in comment 12. We have used different language to describe the possible applicability of this study in other regions because of the overlap with other research. Further, we recommended additional research on the interaction between the different environmental types, as assumed by socio-ecological models. 

Corrections:

                Line 489-490:  Due to the qualitative study design and the recruitment methods, the results of this study may not be generalizable to other populations. … However, similar influential factors may be applicable in other regions as several factors found in this study overlap with previous research into determinants and facilitators and barriers to children’s EBRB in childcare [e.g. 30, 48, 56]. With this study our qualitative knowledge on influential factors has increased on the influential factors in different types of environments. Quantitative studies are needed to evaluate the robustness of these results, in particular on the existence of an interaction between these environments.